# Research on Climate Change and Water Heritage Tourism Based on the Adaptation Theory—A Case Study of the Grand Canal (Beijing Section)

Jiayu Wang , Menghan Wang, Haohan Dou, Mingming Su * , Hangyu Dong and Zhenhua Liu

School of Environment and Natural Resources, Renmin University of China, Beijing 100872, China; jiayuwang@ruc.edu.cn (J.W.); wangmenghan@ruc.edu.cn (M.W.); haohandou@126.com (H.D.); 13011068566@163.com (H.D.); lzh152@ruc.edu.cn (Z.L.)
* Correspondence: smm52@hotmail.com

**Abstract:** Water is at the forefront of climate change and is seen as a major channel through which the effects of climate change are felt. The function of water heritage is closely related to the water bodies on which it depends. Under climate change, the conservation and tourism uses of water heritage resources are facing impacts and challenges. Taking the Beijing Section of the Grand Canal of China as a case, this research applied the adaptation theory to explore the impacts of climate change on heritage tourism of the section of the Grand Canal in Beijing. It was identified that changes in the temperature and the precipitation formed climate-related stimuli to tourism along the Canal from 2012 to 2021 in Beijing. Second, from the supply side of tourism, policies were formulated at a national or municipal level to respond to the changing climate and its impacts on the Canal and its tourism uses. Natural-based solutions (NbS) have been applied to rehabilitate the ecosystem of the Canal, contributing to the enhanced tourism landscape, and providing opportunities for ecological education. Third, from the demand side, high tourism participation along the Canal was examined during the high-temperature years. Meanwhile, the increasing tourist needs for water spaces and activities were observed with evident seasonal patterns. Accordingly, suggestions for climate adaptation of the Grand Canal from a tourism perspective were proposed. For heritage conservation, actions of ecological restoration and monitoring should be further implemented. To assist in the climate adaptation and sustainable development of Grand Canal tourism, suggestions are proposed to enhance the overall tourism planning, increase water accessibility, and heritage interpretation for tourists.

**Keywords:** water heritage tourism; climate change; adaptation theory; the Grand Canal; tourism supply and demand



## 1. Introduction

Climate change has been considered the most significant global environmental issue faced by humanity to date. It has been well established that climate change is a major cause of critical risks to natural and social systems [1]. Undoubtedly, any changes can create opportunities and threats to activities in a region, including tourism [2,3]. Indeed, tourism is both climate-dependent and climate-sensitive [4]. Thus, a better understanding of the interactions between tourism and climate change would be critical to the sustainability and viability of tourism [4,5]. Studies have been conducted to examine the impacts of climate change on tourism. From the supply side of tourism, the changing climate had impacts on the destination product supply, the energy choices, the service levels and prices, the strategic choices, and the policy responses [6,7]. From the demand side, existing studies suggested that climate change affected the psychological perception and behavioral responses of tourists in many aspects such as travel motivation, destination choice, travel experience, travel safety, tourist psychology, and behavior and tourism flow [2,8,9].

Heritage, cultural or natural, is an important resource for tourism, which could enhance the value recognition of heritage resources, enhance the public awareness for heritage conservation, and promote the socio-economic development of the heritage areas [10,11]. Both heritage conservation and tourism development are also intricately linked to climate change. Previous studies identified that climate change would lead to heritage decay in terms of physical, chemical, and biological perspectives [12,13]. Among various types of heritage, water heritage is heavily influenced by climate change. Changes in the climate elements, such as temperature, rainfall, water vapor concentration, sunlight duration, and radiation intensity, may directly or indirectly impact the river or lake runoff, water self-purification capacity, hydrological processes, pollutant migration, and transformation [14–16]. Furthermore, an existing study pointed out that contemporary water heritage had produced irreversible hydrological changes due to natural and climate change impacts. The heritage and cultures that arose from the human–water nexus were thus affected [17]. In addition, the impacts of climate change on water heritage tourism are highly uncertain across geographies and conditions. For example, scholars based on a global perspective argued that tourism and recreation would continue to promote access to water environments under continued global warming [18]. Conversely, the case of the Atlantic coast also provided a suggestion by econometrically speculating that extreme heat significantly reduces recreational participation [19].

Various forms of water heritage have been recognized and inscribed on the World Heritage List, such as canals, landscapes associated with lakes and rivers, hydraulic techniques, and artificial waters as part of the landscape of monumental complexes. They not only record the uses of water by human beings in history, but also witness the process of civilization and social construction that are inseparable from water resources [18]. Beyond other values, water heritage has the potential to serve as an important tourism space to meet the hydrophilic needs of people. However, from reviewing the existing literature, few studies were found to have explored the synergistic solutions between heritage conservation and tourism utilization under climate change.

Adaptation is considered to be an important solution to respond to climate change. Adaptation theory came from the evolutionary theory put forward by Darwin in 1859. Then, adaptation was applied to other fields through interpretation and development, including tourism [20] and heritage [21]. A framework for climate change adaptation was proposed [22], specifying three elements as adaptation to what, who or what adapts, and how adaptation occurs (as seen as Figure 1). The first element refers to climate-related stimuli, which could be interpreted as the natural aspects of climatic conditions or ecological effects or human impacts due to the climate. The second element is explained as a system. Any analysis of adaptation requires a systematic theme and a definition or description of boundaries, distinguishing who acts and modifying what is also to be embodied. The third element relates to the process of adaptation and the resulting forms of adaptation, which are both microscopic and specific.

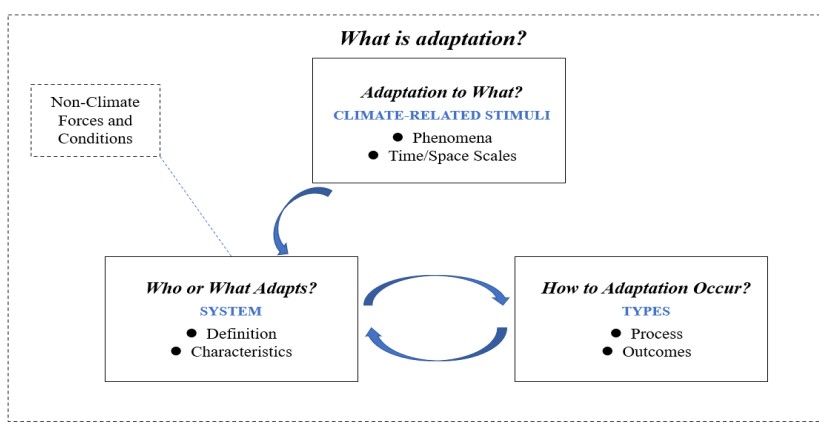

**Figure 1.** Adaptation theory to climate change [12] (Drawn by authors).

As a climate-sensitive industry, tourism research on climate change adaptation could date back to the 1960s [23]. Compared with the frameworks based on business, consumer, destinations, policy, and sustainability [7], the adaptation theory could provide a systematical and dynamic framework to examine the responses of tourism in the changing climate. In addition, three types of key stakeholders in tourism, including policy-makers, tourism operators, and tourists, were included in this study to reflect both the demand and supply of the tourism system.

In this context, this research focuses on the Grand Canal Beijing section, an important water heritage site designated as a UNESCO World Heritage [24]. Drawing on the framework of the adaptation theory (as displayed in Figure 1), the following research questions were formulated to understand the interactions occurring between climate change and canal tourism. First, what changes have occurred in Beijing's climate over the past 10 years and how has it affected tourism? Second, what actions have been taken to adapt to climate change from the supply side of canal tourism? Third, how does the tourism demand adapt to the changing climate? Theoretically, this study is one of the first to explore water heritage tourism and climate change adaptation. Efforts have been made to construct a framework for the utilization and conservation of water heritage tourism based on the supply and demand, which is a supplement to the existing literature. Practically, this study provides explanatory evidence through empirical research. The proposed Grand Canal tourism climate adaptation scheme can provide an accurate reference for the sustainable development of water heritage tourism.

## 2. Materials and Methods

### 2.1. Study Area

In 2014, the Grand Canal was listed in the UNESCO World Heritage as a mixed (cultural and natural) heritage site [24]. With a total length of more than 2000 km, the Grand Canal links five major basins in China from Beijing in the north, to Zhejiang in the south. It is the longest canal in the world with a history of more than 2500 years. Both in ancient times and today, it has outstanding engineering, socio-economic and political values in the development of the country and the life of the people. Beijing is the northern terminus of the Grand Canal. In the Beijing section, the 82 km long canal runs through six districts: Changping, Haidian, Xicheng, Dongcheng, Chaoyang, and Tongzhou (as seen in Figure 2).

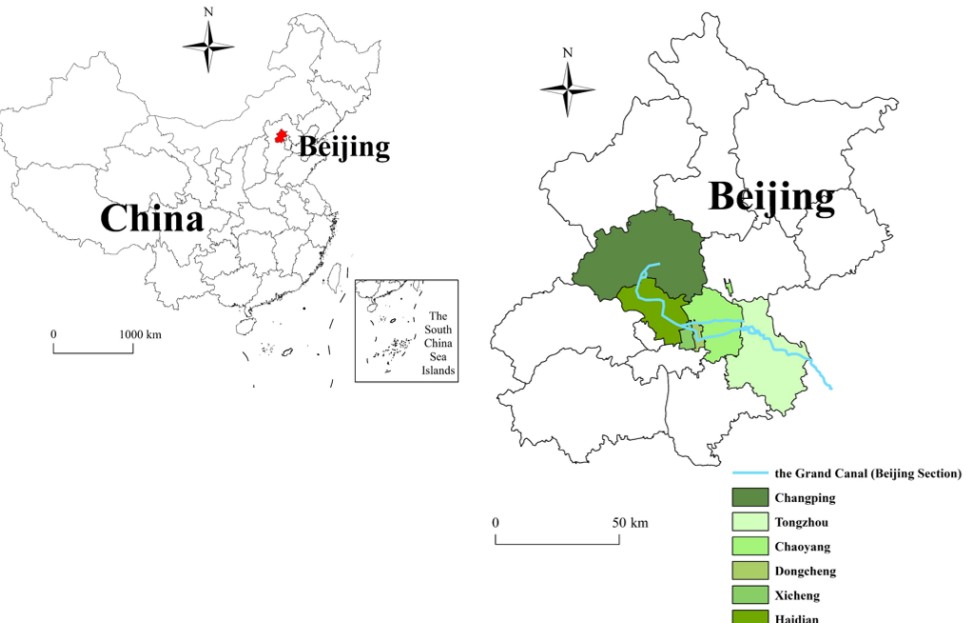

**Figure 2.** Location map of the Grand Canal (Beijing Section) (Drawn by authors).

The Grand Canal connects the historical and cultural resources along the route. After being authorized as a UNESCO World Heritage Site, the integration of resources and the increased visibility have brought new developmental opportunities for tourism of the Grand Canal and the area which it flows through [25]. In Beijing, there are many well-known tourist attractions along the Grand Canal (as seen in Figure 3), such as Shichahai, Tongzhou Grand Canal Forest Park, and Yuhe Park. The water bodies within these spots undertaking tourism functions are integral parts of the Grand Canal (Beijing section).

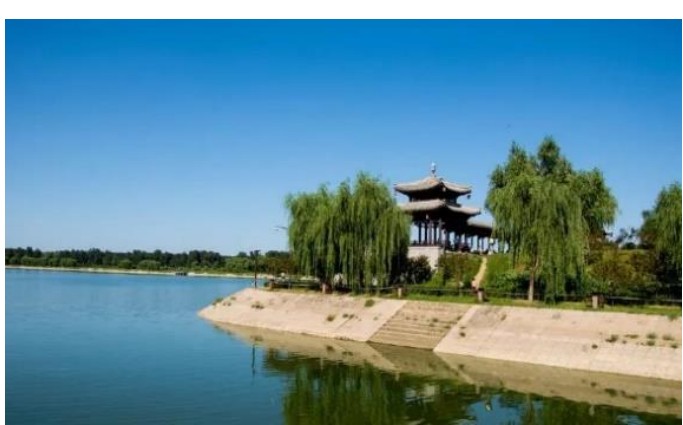 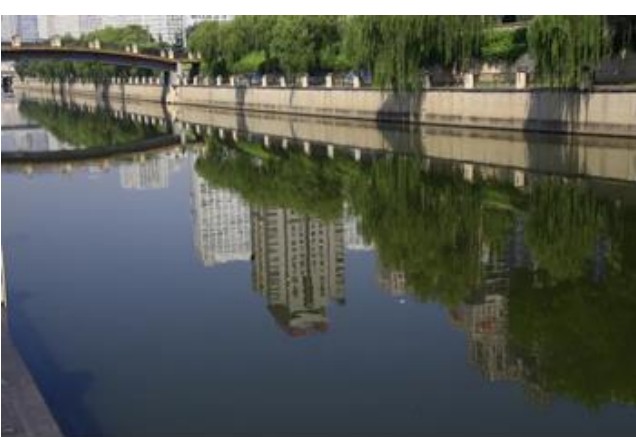

**Figure 3.** The tourist attractions along the Grand Canal (Beijing section); (Photos by authors).

Climate change imposes negative impacts on the human and natural world in various ways, in which water-related issues are particularly pronounced [26]. Thus, the conservation of the Grand Canal is also challenged by climate change. Therefore, increasing its resilience to climate change is among the objectives of heritage conservation and tourism development for the Grand Canal.

*2.2. Data Collection and Processing*

According to the adaptation theory (Figure 1), corresponding research methods were designed. First, statistical analysis was conducted to study the climate characteristics of Beijing and the correlations between the tourism and the climate over 10 years. Theoretically, a decade was considered to be an appropriate time scale for measuring climate change and adaptation [22]. Thus, taking 1 January 2012, to 31 December 2021, as a time slice, panel data on temperature, precipitation, and tourist volumes from 2012–2021 were collected through Python software from the Beijing Meteorological Bureau and China Meteorological Data Sharing Service System, and the Beijing Municipal Bureau of Culture and Tourism, respectively. Then, the visual representation of the monthly temperature and precipitation data was carried out and then statistically analyzed. Additionally, tourism flow was considered to be an important indicator for exploring tourism and climate change [27]. Based several previous studies [28,29], a regression model was generated using the mean monthly temperature and mean monthly precipitation, and the monthly tourist volume.

Second, the research team conducted fieldworks along the Canal in Beijing from 8 February to 10 March, 6 April to 8 April, 3 December, 11 December 2022, and 18 March 2023, respectively. Important sections and areas of the Grand Canal (Beijing section), such as the Liangma River, Shichahai, Yuyuantan Park, and Tongzhou Grand Canal Forest Park, were included in fieldwork. The landscape, facilities, and environmental constructions along the Canal to adapt to climate change were the main objects of observation. In addition, five casual tourist interviews were conducted in this process to understand their use patterns and their relation to the changing climate, which supported the quantitative analysis.

Third, content analysis was conducted based on user-generated content (UGC) to understand tourism demand. Thus, web travelogues and reviews about various attractions of the Grand Canal (Beijing section) from 1 January 2012 to 31 December 2021, were

collected through Python from top online travel platforms in China as Ctrip and Mafengwo. It should be noted that the number of UGC was related to the popularity and the usage of travel social media. However, by observing it together with climate indicators, some results were generated. Finally, a total of 293,841 words were obtained. The word frequency analysis and theme analysis were conducted. To further explore the connections between the temperature and the level of tourism participation, direct quotations were extracted from online reviews, such as "summer" and "winter".

Data were collected and analyzed first in Chinese and then translated into English. To ensure the validity of the research, the results were reviewed and cross-checked.

## 3. Results

### 3.1. The Climate Change in Beijing and the Stimuli to Tourism

3.1.1. The Changing Climate in Beijing from 2012–2021

Temperature and precipitation are considered to be the most direct indicators for studying climate change [30]. Beijing has a temperate monsoon climate, and the analyzed climate indicators also showed significant seasonal differences corresponding to it. Figure 4 shows the monthly temperature and precipitation in Beijing from 2012 to 2021.

For temperature, the average annual (from 2012–2021) temperature was 13.2 °C. During the 10 years, the temperature showed an increasing trend, as was evidenced by the average annual temperature after 2014. Of note, the highest temperature was recorded in 2017, reaching 14.2 °C. In addition, changes were also occurring in the annual seasonal temperature differences between the four seasons. In particular, it was the largest in winter, with the highest annual average temperature of 1.1 °C, and the lowest annual average temperature recorded was −4.1 °C, with a difference of 5.2 °C between the two. The difference between the highest annual average temperature and the lowest annual average temperature in spring, summer, and autumn is 4.4 °C, 4.3 °C, and 3.4 °C, respectively.

The average annual precipitation was 554.64 mm from 2012–2021. From Figure 4, it can be found that the annual precipitation in Beijing was extremely uneven. The annual precipitation was greater than the multi-year average precipitation in five years, namely 2012, 2013, 2016, 2017, and 2021, respectively, among which the highest precipitation recorded was 733.2 mm in 2012 and caused flooding. The annual precipitation was found to be less than the multi-year average precipitation in five years, namely 2014, 2015, 2018, 2019, and 2020, respectively, of which 2019 was the year with the least precipitation. From 2012 to 2021, the average precipitation in Beijing varied greatly from year to year in all seasons. Beijing's summer (from June to August) was the rainy season, with precipitation accounting for more than 70–75% of the year, while winter had very little rain and snow, with precipitation accounting for only 2% of the year, spring was the next least, accounting for 8–10% of the year, and autumn accounted for 13–16% of the year.

3.1.2. The Stimuli of Climate to Tourism

As shown in Figure 5, changes in the temperatures and levels of precipitation in Beijing's summer and winter from 2012 to 2021 can be described. In summer, June and July were the months worthy of attention. Although the hottest month was not June, the most extreme maximum temperatures occurred in June. July was the summer peak, with an average temperature close to 26 °C. The high temperature was stable, and the temperature difference between day and night was small. Summer precipitation accounts for 70% of the annual amount, mostly in the form of heavy rainfall. Plain flooding was one of the main natural disasters in Beijing in the summer. Meanwhile, winter was found to be cold and long in duration. Winter precipitation accounted for 2% of the annual precipitation, and there was often no precipitation (snow) lasting for more than one month.

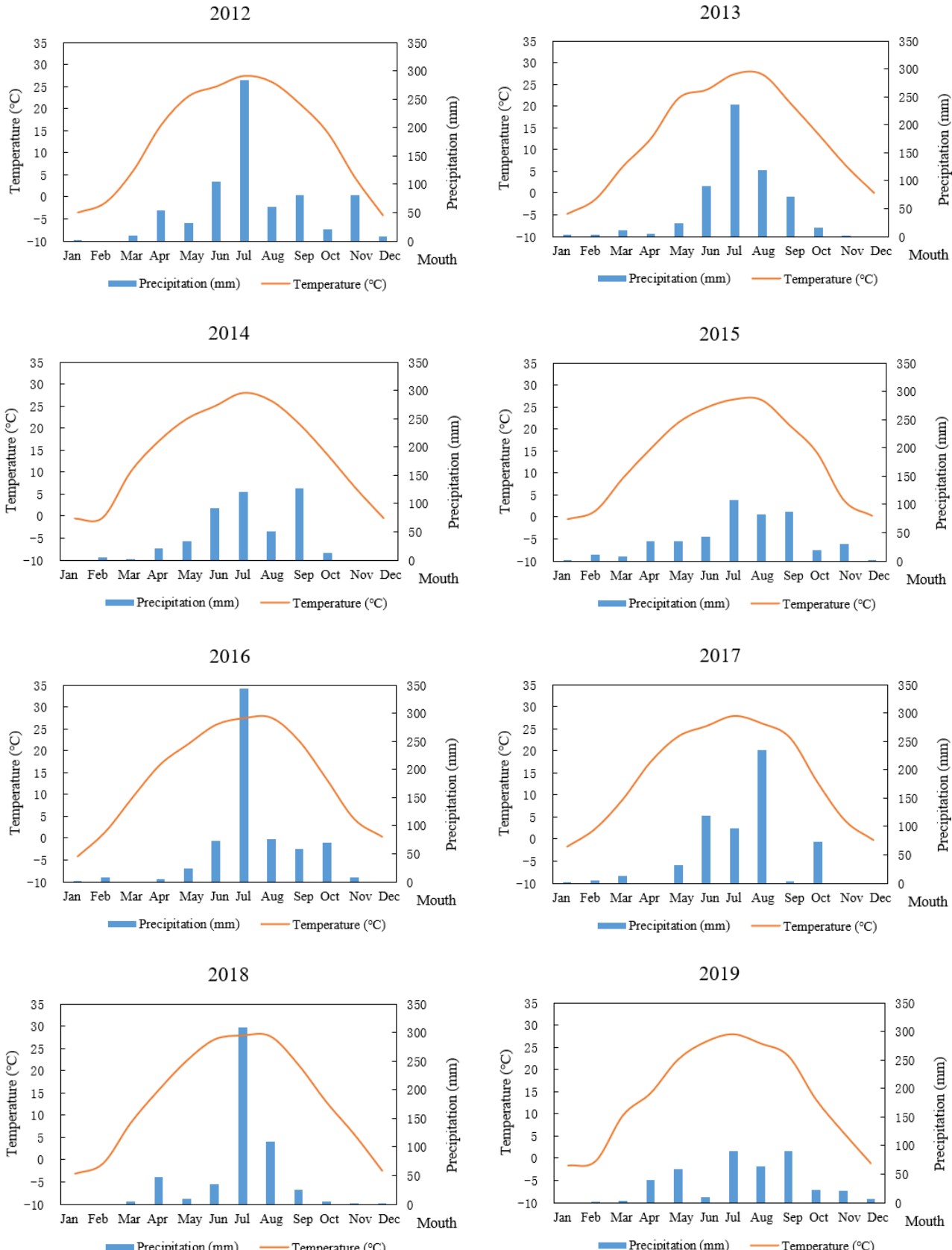

**Figure 4.** *Cont.*

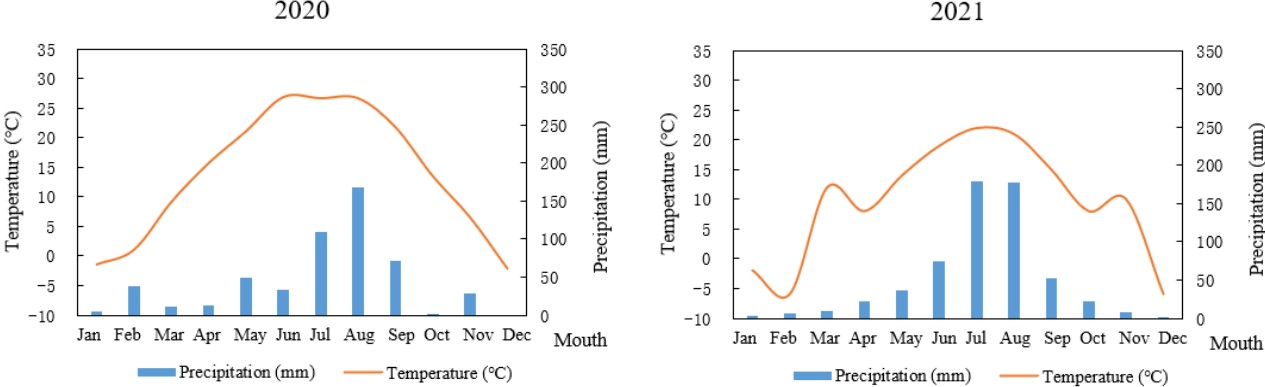

**Figure 4.** Monthly temperature and precipitation in Beijing from 2012–2021.

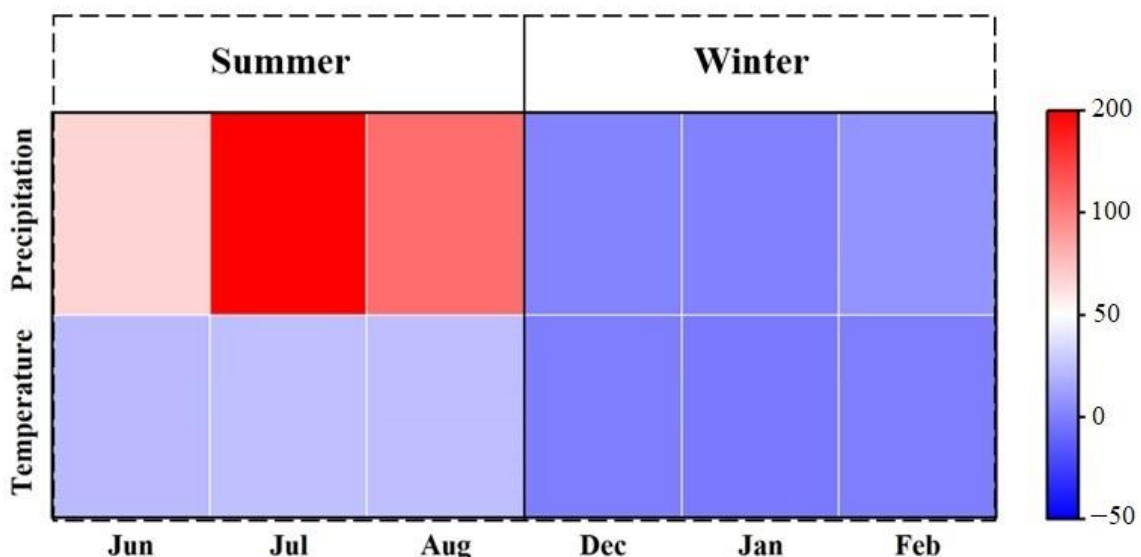

**Figure 5.** Comparisons of the temperature and precipitation between summer and winter.

As shown in Table 1, a trend of increasing tourist volume with increasing temperature, along with a decreasing trend with increasing precipitation in winter and spring, were observed. It would therefore be helpful to balance the intra-year distribution of tourists and flatten the seasonal variation. Correlations between visitor volume and variations in temperature or precipitation were not found to be significant in summer and autumn. However, the results also revealed the correlation between changing climate and tourist volumes, which therefore established a scientific link between climate change and tourism for this research.

**Table 1.** Correlation between climate indicator and visitor volume in different seasons.

|  | $R^2$ | Pearson Value | Sig. (Two-Tailed) |
|---|---|---|---|
| Spring | 0.798 | 0.871 ** | 0.000 |
| Summer | 0.415 | 0.592 | 0.040 |
| Autumn | 0.048 | 0.139 | 0.581 |
| Winter | 0.679 | 0.826 ** | 0.002 |
| Year-round | 0.315 | 0.610 ** | 0.000 |

Note: **. Correlation is significant at the 0.01 level (two-tailed).

*3.2. Supply Side Adaptations to the Changing Climate of the Grand Canal (Beijing Section)*

3.2.1. Policy Responses to Climate Challenges

As shown in Table 2, more specialized policy and strategic actions were developed at the national and Beijing municipal levels from 2012 to 2021. To address the risks of climate change, China's National Program to Address Climate Change and National Adaptation Strategy to Climate Change were released in 2007 and 2013, respectively. In 2022, China's Ministry of Ecology and Environment and 17 other departments jointly issued the National Adaptation Strategy for Climate Change 2035 (NASCC), which referred to the necessity of nature-based solutions to enhance the adaptive capacity of water resources, including the Grand Canal.

**Table 2.** Policies on Grand Canal at the national and municipal levels from 2012–2021.

| Year | Policy | Description | Authorizing Body |
|------|--------|-------------|------------------|
| 2012 | Beijing Flood Control Emergency Plan | To resist flood disasters, Beijing has carried out urban flood control construction, and completed the reinforcement of key flood control channels and embankments such as the Yongding River, Chaobai River and North Grand Canal. | Beijing Municipal Emergency Management Bureau |
| 2012 | Measures for the Administration of the Heritage Protection of the Grand Canal | The department in charge of cultural relics under the State Council is in charge of the overall protection of the Grand Canal heritage site, and cooperates with the departments in charge of the land, environmental protection, transportation, and water conservancy of the State Council to carry out related work within the scope of their respective responsibilities according to law. | Ministry of Culture of the People's Republic of China |
| 2013 | Opinions on strengthening the construction and management of the ecological environment of rivers and lakes (2013–2015) | (1) Accelerate the construction of river and lake greenways; (2) Implement the construction of "two canals and ten rivers" greenways, improve the ecological environment around rivers and lakes, and serve citizens for leisure and fitness. | General Office of Beijing Municipal People's Government |
| 2016 | Beijing 2016 Flood Control Work Plan | Tourism special sub-points perform a good job in the safe transfer of tourists and the safety of scenic spots during flood seasons. | Beijing Municipal People's Government |
| 2017 | Beijing Urban Master Plan (2016–2035) | Promote the construction of the Grand Canal Cultural Belt, the Great Wall Cultural Belt, and the Xishan Yongding River Cultural Belt. | Central Committee of the Communist Party of China |
| 2018 | Protection and Construction Plan of Beijing Grand Canal Cultural Belt | (1) Accelerate the monitoring of the water environment along the Grand Canal to keep abreast of the water quality; (2) Improve the landscape environment quality of the Grand Canal. | Standing Committee of the Beijing Municipal Committee |
| 2019 | Regulatory Detailed Planning of Beijing City Sub-center (Block Level) (2016–2035) | (1) Relying on cultural resources such as the Grand Canal Cultural Belt, create a new window for culture and tourism; (2) Ensure the safety of urban flood prevention and waterlogging; (3) Provide more high-quality and convenient green spaces for leisure and recreation for the people. | Beijing Municipal Commission of Planning and Natural Resources |
| 2019 | "The Great Wall, the Grand Canal, and the Long March National Cultural Park Construction Plan" | (1) Restoring the space environment, giving full play to the restoration of natural ecosystems, controlling soil erosion, and carrying out water pollution prevention and control; (2) Improving tourist routes and connecting important nodes. | General Office of the Central Committee of the Communist Party of China and the General Office of the State Council |

**Table 2.** *Cont.*

| Year | Policy | Description | Authorizing Body |
|---|---|---|---|
| 2020 | Beijing's medium and long-term plan for promoting the construction of a national cultural center (2019–2035) | Make overall plans to promote the construction of the Grand Canal Cultural Belt and build an overall spatial structure that blends historical context and ecological environment. | Beijing Municipal Leading Group for Promoting the Construction of a National Cultural Center |
| 2021 | Construction and Protection Planning of Beijing Grand Canal National Cultural Park | (1) Promote the management of the ecological environment along the Grand Canal, build a waterfront leisure space near the water, and create an ecological and cultural landscape corridor; (2) Strengthen greening and environmental improvement on both sides of the river, and build a flood control system. | Beijing Municipal Development and Reform Commission |

With this background, Beijing launched the Protection and Construction Plan of the Grand Canal Cultural Belt in 2018 [31]. In 2021, Beijing released the Construction and Protection of Grand Canal National Cultural Park, where ecological governance and tourism development were presented as core topics. To cope with extreme events under climate change, the construction of the Grand Canal flood control system was introduced. For example, the Grand Canal was cut off in the first half of the 20th century due to historical evolution, human activities, and climate change. In 2022, the Grand Canal initiated a full replenishment of water, and a replenishment volume greater than 515 million cubic meters allowed the Grand Canal to be fully navigable [32], which provided macro support for the development of the Grand Canal.

From the ecological perspective, water system management and ecological reconstruction along the canal played an important role in climate change adaptation and biodiversity conservation. Moreover, ecological improvements and the creation of river landscapes and leisure spaces (Figures 6 and 7) could enrich the tourism supply and increase the tourism potential along the canal. From the tourism perspective, the importance of the Grand Canal as leisure space to enhance resident well-being has been emphasized. Moreover, the canal was considered a critical carrier of the national identity, which needed to be constructed [33].

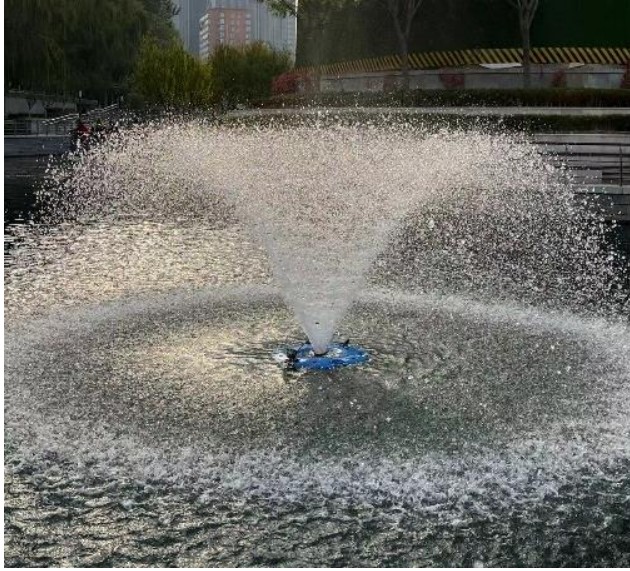 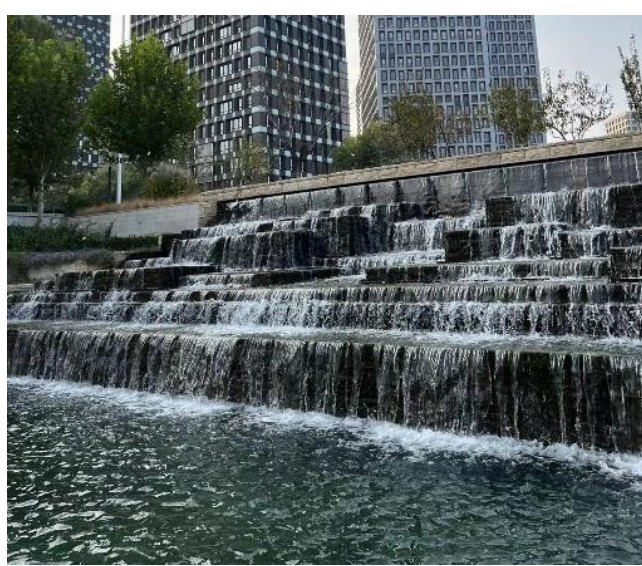

**Figure 6.** The integration of water management and tourism landscape in Liangmahe; (Photos by authors).

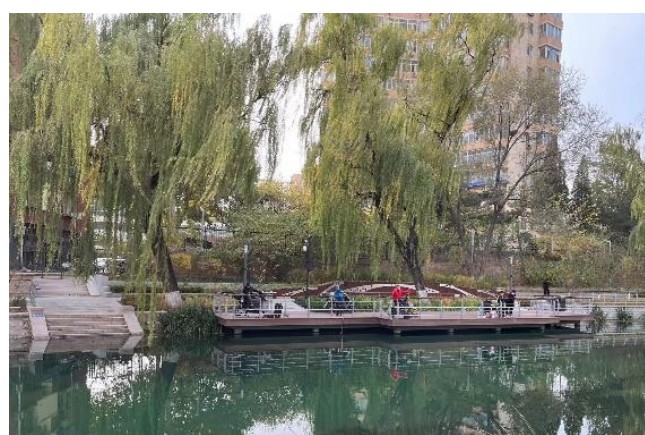
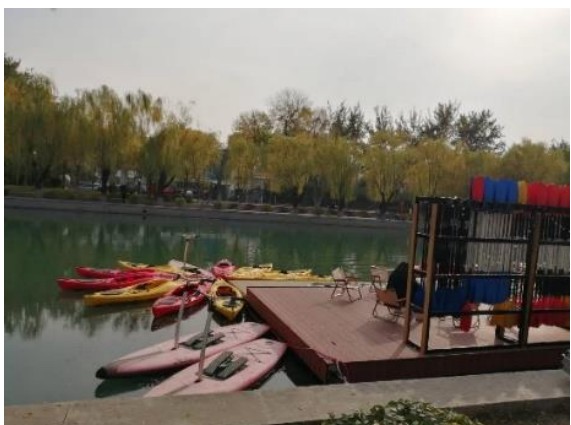

**Figure 7.** The construction of the waterfront space and water-related tourism supply; (Photos by authors).

3.2.2. Existing Practices Based on Nature-Based Solutions (NbS)

As mentioned above, nature-based solutions (NbS) were seen as an important adaptation option for climate change and canal management. NbS were defined as actions to conserve, sustainably use, and restore natural or altered ecosystems, which can effectively and adaptively respond to the challenges facing our society today, while enriching human well-being and biodiversity [34,35].

It was observed that NbS were applied along the Grand Canal (Beijing section). In areas close to the canal, floodplains and grassed ditches were constructed (as shown in the left of Figure 8), which could act as the buffer zone during heavy rainfalls, which appeared more often in recent decades, and could thereby support flood management in Beijing. Meanwhile, the rehabilitation of the canal could optimize the canal landscape and enhance its tourism value. In riparian areas, green corridors built with biological measures provided habitats for local birds and insects. In addition to providing additional tourist space, these corridors could contribute to carbon reduction by sequestering carbon dioxide and releasing oxygen. Moreover, many interactive interpretative facilities were designed along the canal. Content on wildlife conservation and climate change was explained in entertaining formats such as the rotating rubes in Figure 8 (on the right).

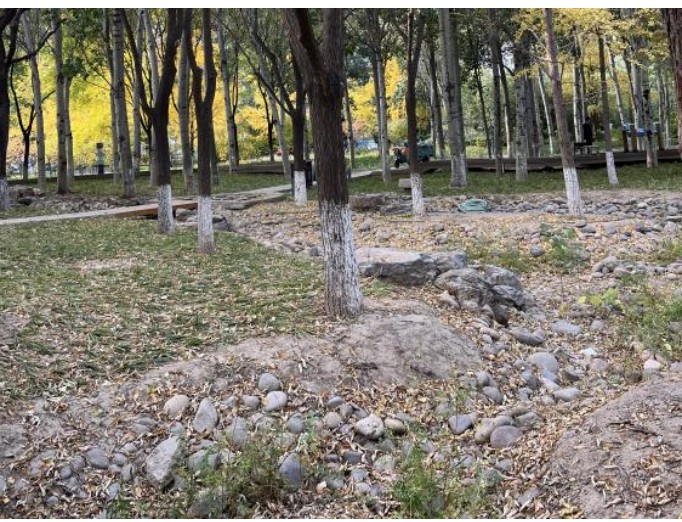
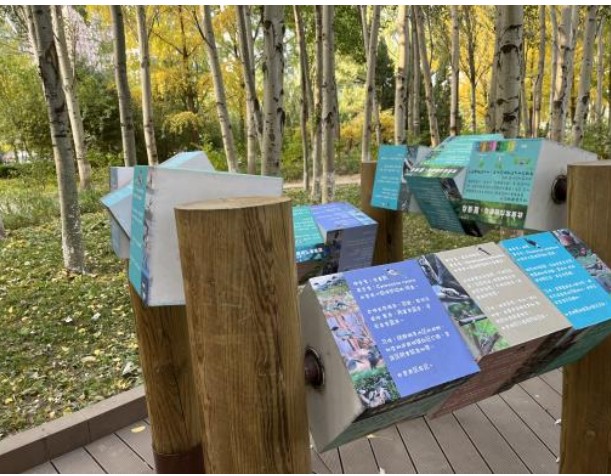

**Figure 8.** The NbS measures and tourism interpretative facilities near the Grand Canal; (Photos by authors).

It could be considered that the implementation of NbS was an effective approach in the policy process. In this research, the supply of canal tourism under climate change was elucidated in a specific way. The hermeneutic approach was similar to extracting a key frame from a series of shots. It was found that NbS measures could promote the conservation of the canal heritage and enhance its ability to respond to climate change. Moreover, NbS also enriched the tourism supply by extending the resources for tourism and generating ecological education effects for the tourists.

### 3.3. Demand Side Adaptations to the Changing Climate

3.3.1. Climate-Related Tourism Participation

Initially, to match the link between tourism participation and the climate, the collected data of UGC and the climate panel were analyzed by year time slices. As shown in Figure 9, the number of UGCs between 2012 and 2021 was fluctuating. From 2012–2021, the average annual temperature in Beijing was 13.2 °C. Only 2014 and 2017 demonstrated average temperatures higher than the 10 year average of 13.2 °C. Correspondingly, two peaks in the number of UGCs were also examined in 2014 and 2017. Even considering that the number of UGC increases year by year with the increasing uses of social media, such a correlation is still indicative of a higher tourism participation along the Grand Canal in Beijing in the hotter years.

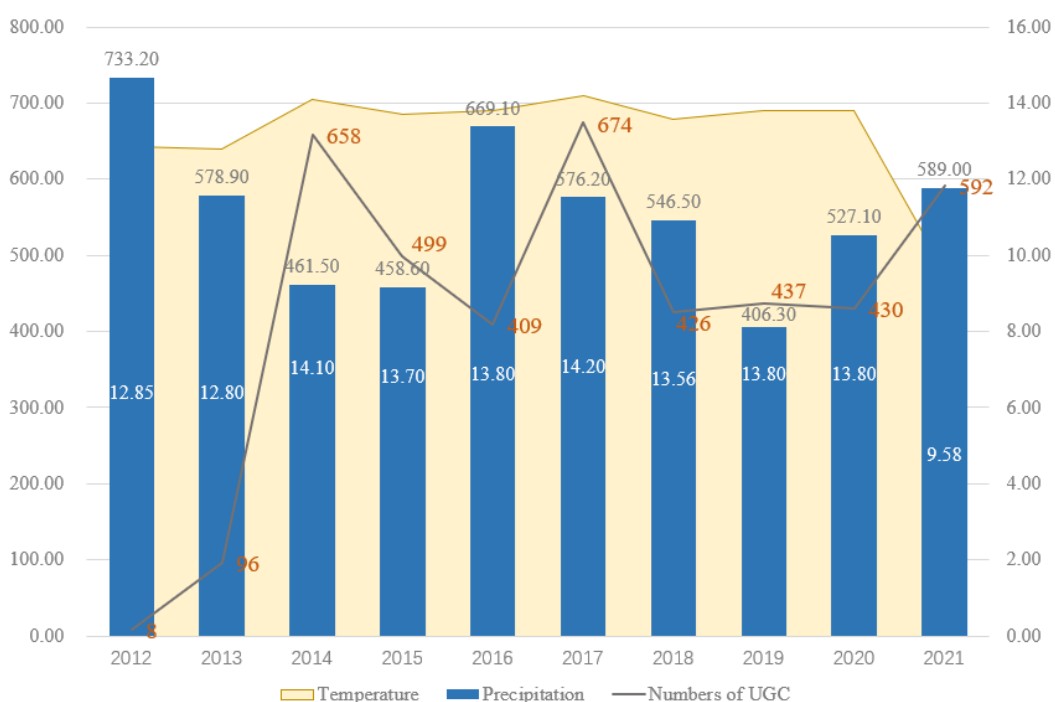

**Figure 9.** The temperature, precipitation, and tourist volumes in Beijing from 2012–2021.

To further explore the connections between the temperature and the level of tourism participation, direct quotations were extracted from online reviews. According to the climatic characteristics of Beijing, high temperatures occurred during the summer. Therefore, the UGC of 2014 and 2017 were extracted with "summer" as the main filtering condition. Co-occurrence of "summer" and "coolness" was found among canal tourists, and water-related tourism participation was featured.

*"The lake surface is quite large. Although it is summer and the temperature is very high, it is still cool to stroll along the Kunming Lake with the willows leaning against it" (3 August 2017, Kunminghu, Summer Palace)*

*"It's right next to Houhai, a great place to cool off in summer evenings. There's a bar nearby. People who enjoy excitement can go drink and chat, while those who enjoy quietness can stroll around Shichahai. It's cool, but there are more mosquitoes"* (18 July 2014, Shichahai)

*"Take a walk along the river, and at night, the wind blows by, making it cool and relaxing in summer"* (30 June 2014, Tongzhou Forest Park)

*"During the day, there are not many people here, and it is very quiet. Occasionally, elderly people are fishing by the lake, which is very pleasant. In summer, this is also a place to enjoy the lotus in Beijing"* (5 April 2017, Shichahai)

*"Boating on the lake is particularly comfortable. Looking at the scenery on the shore, it feels very cool in summer"* (30 July 2014, Kunminghu, Summer Palace)

Thus, a clearer connection between the changing climate and canal tourism was constructed. With the canal offering water experiences and activities that could mitigate the negative experiences from the rising temperatures induced by the changing climate, higher demands for canal tourism were examined, especially in the summer.

3.3.2. The Increase of Water-Based Tourism Activities along the Canal

To further verify the hydrophilic needs of tourists, the word frequency analysis was conducted based on the UGC from 2012–2021. Meanwhile, the climate data showed the seasonal characteristics of climate change generated in Beijing, which provided a basis for exploring the perceptions and behaviors of the Grand Canal (Beijing section) tourists by seasonal divisions.

The high-frequency words could reflect the overall perceptions of the tourists in different seasons. Accordingly, the top 30 high-frequency words were shown in summer and winter. As shown in Figure 10, tourists' perceptions of the Grand Canal (Beijing section) in summer focused on "boating", "swimming", "wakeboarding", "sailing", "sailing", and "paddle boarding". This indicated that the Grand Canal (Beijing section) was the place for tourists to engage in water-based activities during the summer with high temperatures. While in winter, "ice" becomes the core resource of recreational uses with "ice skating", "fishing", "winter swimming", and "ice rink" having been listed. An evident seasonal pattern in tourist participation in the Grand Canal (Beijing section) was identified as water-based activities in summer and ice-based activities in winter, and both were deemed to be prone to climatic influences.

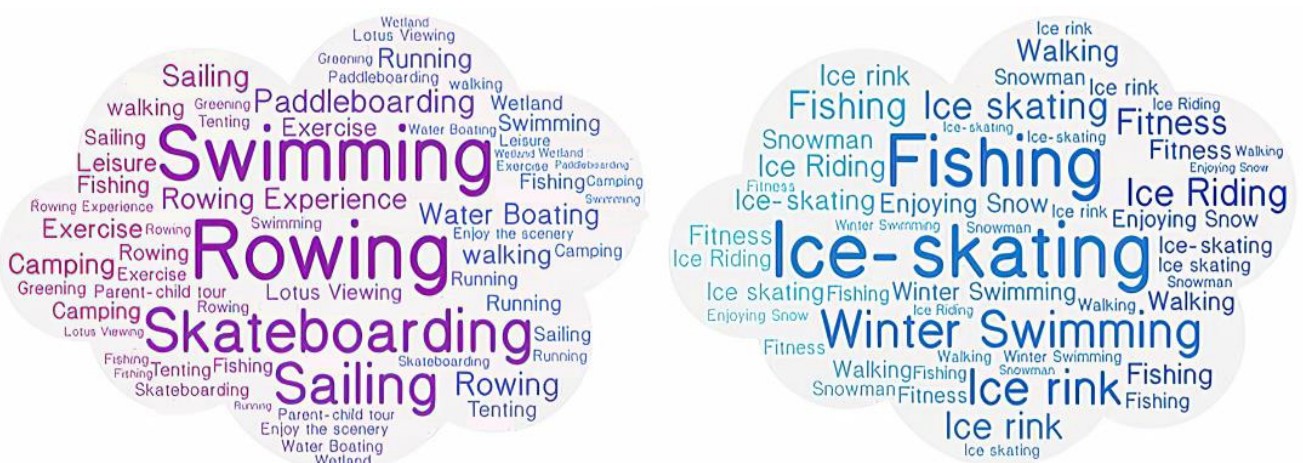

**Figure 10.** Comparison of tourists' perception in summer (**left**) and winter (**right**).

Similarly, the field research along the canal corresponded with the seasonal patterns illustrated by the online reviews. Swimming and fishing were primarily engaged by elders of both genders while fishing was favored among males. As one elder man in his 90s

described: *"I ride my bike to the Canal almost every day, except in bad weather. I like fishing . . . Oh, yes, there are fish in the Canal, small fish. I catch fish and then release them. It is a kind of routine of my life and it is good for my health."*.

Additionally, there were also emerging uses of the canal by the younger generation through more active water activities such as sailboarding. As a female interviewee with young children expressed: *"These summers have been really hot and dry, so taking advantage of the canal, we brought sailboards from Taobao (the popular online shopping platform in China) and play sailboards quite often in the summertime. It is cool along the Canal. My children enjoy it very much and it is good for their health too!"*.

These interviews also indicated influences of the pandemic in the past three years, which has encouraged within-Beijing recreational activities when trips outside Beijing were restricted. Along the Grand Canal (Beijing section), tourists and residents enjoyed the water with sketching, swimming, and fishing (as seen in Figure 11).

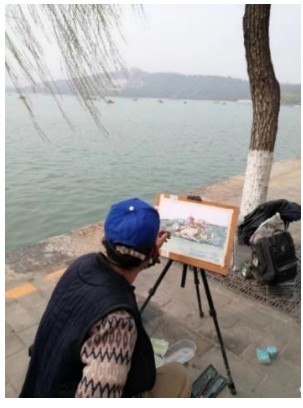 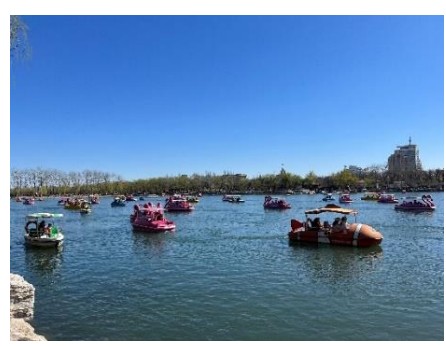 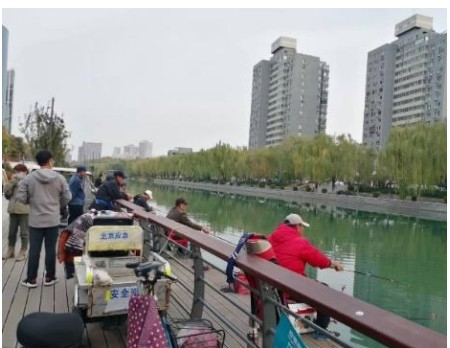

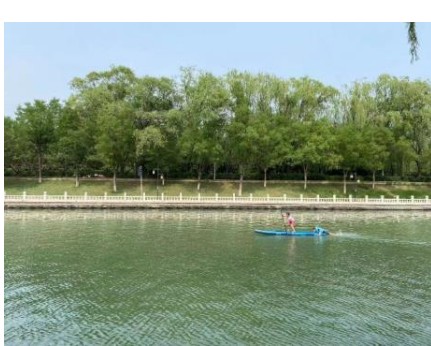 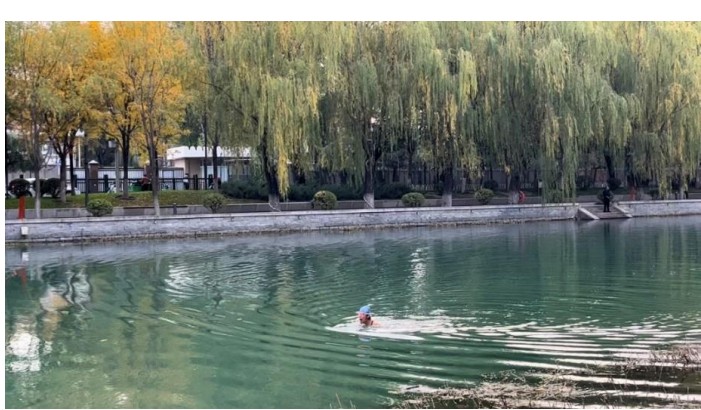

**Figure 11.** Tourist participation in water-related activities along (and in) the Grand Canal (Beijing section); (Photos by authors).

Thus, in responding to the changing climate and the pandemic, the Grand Canal (Beijing section) has been playing an increasingly important role in supporting the increasing demands of water-related activities among residents and tourists in Beijing.

## 4. Discussion and Implications

### 4.1. General Discussions

Previous studies have revealed that climate change is already impacting heritage resources [12,13]. With tourism as a typical use of these heritage resources, it is important to therefore understand the interactive relations between climate change and heritage tourism. Taking the Beijing section of the Grand Canal as a case, this research examined the adaptation to climate change on the supply and demand side of the Grand Canal tourism, and the adaptation mechanism was developed (as shown in Figure 12).

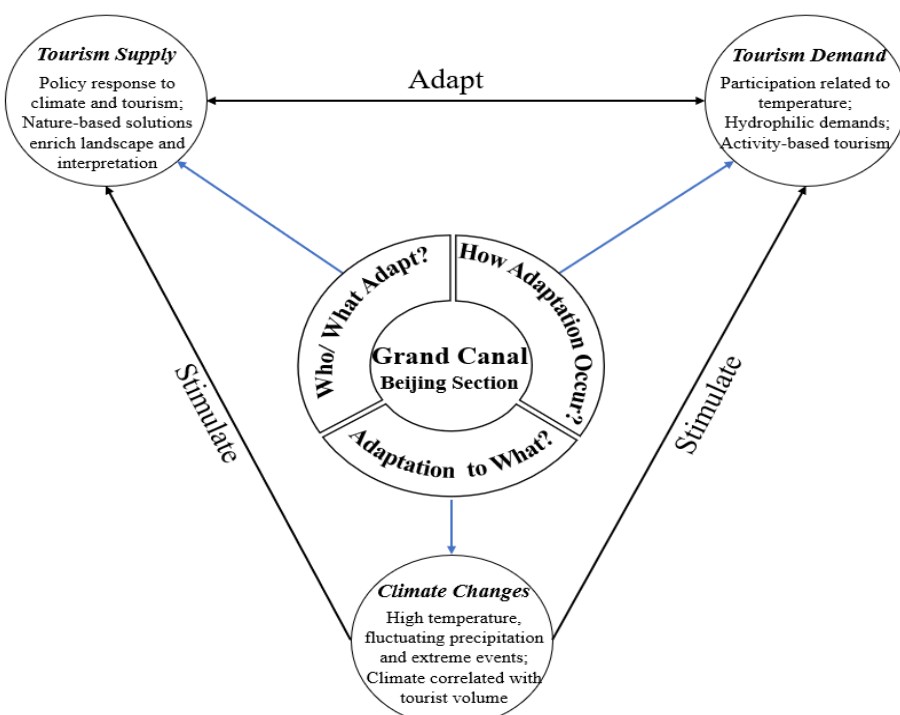

**Figure 12.** Adaptation mechanism for tourism of the Grand Canal (Beijing section) under climate change.

First, over the past 10 years, climate change in Beijing was reflected in high-temperature years, seasonal differences in precipitation, and extreme weather events. The direct stimulus of the changing climate on tourism was expressed in the change in tourist volume. Through the correlation analysis between climate indicators and tourist volume, a high correlation (0.871 ** and 0.826 **, respectively) between the two in spring and winter was found. These findings illustrated that there were stimuli to tourism originating from a changing climate, while also validating the results of previous studies on the impact of climate change on tourism flow [27]. Furthermore, it provided a basis for this research to study water heritage tourism under climate change in Beijing.

Second, actions of adaptation occurred in the tourism supply that were stimulated by climate change. In this research, policies for climate change adaptation and water heritage conservation at the national and regional levels were articulated in China. Policies and strategies were focused on the ecological and cultural dimensions of the Grand Canal in an umbrella form, which provided a roadmap for actions on the climate change response as well as tourism development for the Grand Canal. Additionally, considered as one of the most promising approaches to climate challenges [35], NbS were engaged as key actions of adaptation. Through the observation during the fieldwork, measures rooted in NbS, such as green corridors, rain gardens, or flood plains along the canal, have improved the ecological quality and the climate regulation capacity of the canal. Meanwhile, it also enriched the supply of the tourism landscape and the waterfront space and is thereby shaping the perception of the tourism demander.

Third, climate change was found to stimulate the demand of tourism, which was leading to adaptation. From searching for the annual patterns of the UGC and temperature indicators, it was be found that high-temperature years led to more tourism participation. Moreover, the co-occurrence of "summer" and "coolness" was found by extracting the UGC in high-temperature years, which showed that the tourists adapted to a high temperature by participating in Grand Canal tourism. This confirmed the findings of previous studies that tourism could promote access to aquatic environments in the face of continued climate change [18]. Furthermore, the word cloud map showed that tourists form a tourism perception with "water" and "ice" as the kernel in summer and winter,

respectively. These phenomena were also verified by interviews and field observations. Consequently, a hydrophilic demand was generated under the influence of climate change, whose specific performance was the increase in activity-based tourism related to water along the Grand Canal.

*4.2. Theoretical and Managerial Implications*

Theoretically, this research provides several contributions. First, previous studies have treated tourism and heritage as two separate entities to establish the relationships with climate change. In contrast, this research treated water heritage and tourism as a system to study their adaptation to climate change, which filled the research gap. Second, previous research on heritage, tourism, and climate change was mostly non-empirical. However, this research was an enrichment of the relevant studies by developing a study through an empirical case. Third, the integration of the climatological adaptation theory to study heritage tourism was advanced. This research extended the application area of the adaptation framework and validated its rationality.

Managerial implications to respond to the heritage conservation and tourism utilization under climate change were then generated for the Grand Canal Beijing. First, the sustainability of the watershed was the basis for climate change adaptation and heritage tourism. Continuous water replenishment according to the changing climate is necessary, which could ensure the water quantity of the canal to sustain its ecological and recreational functions. Additionally, it is of urgent need to develop a risk management plan to deal with extreme weather events. Potential negative environmental impacts of tourism should be attended to through water quality monitoring and tourist behavior management.

Moreover, as a linear heritage, it is necessary to shift the current development focus on special spots, to a whole area development mode that links the linear water body and the radiating area. Themed and diverse touristic routes should be designed through improving the connectivity between the different sections of the canal. Moreover, the conservation and tourism uses of the Grand Canal should be included in overall urban planning to enhance the connectivity of the canal to the wider urban space in Beijing. Meanwhile, heritage interpretation should be diversified both in contents and in the modes [36], so that integral values of the canal can be conveyed through tourism, which could not only enhance the tourism experience, but also support the canal conservation.

Finally, to respond to the increasing demands of tourists and residents on water space and activities, it is necessary to increase the leisure spaces along the waterfront areas along the Grand Canal Beijing. Furthermore, considering the increasing demand for a variety of water-based activities, water sports operations by professional sports companies should be facilitated to cater to the increasing need for water activities, particularly in the summer. Safety insurance infrastructures and measures also need to be developed for water activities or sports, such as swimming or boating.

**5. Conclusions**

Despite the increasing number of academic papers on climate change and tourism, water heritage tourism under climate change is still an under-researched topic. Moreover, the adaptation theory, as an important coping strategy for climate change, requires additional attention in tourism research, which was considered in this study. With the Grand Canal Beijing section as an example, this study draws data from multiple sources as guided by the adaptation theory. Insights for water heritage tourism and sustainable development under climate change were provided.

As a water heritage site with outstanding cultural and natural values, the Grand Canal Beijing Section serves as an important leisure and tourism space for tourists and residents and is vulnerable to climatic impacts. On a decade scale, evidence for climatic changes in Beijing has been identified, particularly in precipitation and temperature, which would stimulate changes in tourism use patterns. Such changes triggered actions on the canal management and shifted demands for tourism uses along the canal. Accordingly, measures

were proposed to promote the conservation of the Grand Canal, enhance its contribution to urban development, and promote viable canal tourism.

In contrast to previous studies that focused on the negative impacts of climate change, this study took a more neutral perspective and explored how a changing climate affects water heritage tourism with a good applicability. The resilience of water heritage enables it to adapt to constantly changing situations compared with other types of heritage. Meanwhile, the natural and cultural properties of water heritage have demonstrated a high potential to meet people's needs for water-related activities under climate change. Thus, this study provides a new perspective to view the impacts of climate change positively and to take action.

This study strove to draw a variety of data sources to illustrate the intricate relationships between heritage tourism and climatic changes. However, there were several limitations. First, as a linear heritage, there are multiple spots along the canal with different recreational and tourism uses, which complicated the analysis of its tourism demands. Second, climate change is among the most complex and intricate environmental issues globally. Data on temperature and precipitation were used as indicators due to data availability, and which do not fully reflect the status of climate change. In the future, more indicators such as greenhouse gas concentrations, the rise of sea level, or the number of extreme weather events should be included. In addition, besides tourism demand and activity participation, the impacts of climate change on the environmental behavior as well as the cultural identity of tourists and local residents are worthy of further study.

**Author Contributions:** J.W. and M.S. drafted the manuscript, M.W. conducted the data analysis and visualization, H.D. (Haohan Dou) participated in the fieldwork and carried out part of the research, M.S. and J.W. developed the research framework, and H.D. (Hangyu Dong) and Z.L. participated in the fieldwork. All authors have read and agreed to the published version of the manuscript.

**Funding:** This work was supported by the Beijing Philosophy and Social Sciences Project (21GLB026).

**Institutional Review Board Statement:** Not applicable.

**Informed Consent Statement:** Informed consent was obtained from all subjects involved in the study.

**Data Availability Statement:** The data are available from the corresponding author upon reasonable request.

**Acknowledgments:** Sincere thanks to all the interviewees for their participation in the research.

**Conflicts of Interest:** The authors declare no conflict of interest.

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
