# Peer review of "Research on Climate Change and Water Heritage Tourism Based on the Adaptation Theory—A Case Study of the Grand Canal (Beijing Section)"

_sustainability, doi:10.3390/su15097630_

Round 1

Reviewer 1 Report

1. Highlight the scientific value of the paper in the abstract.

2. Clarify the novelty of the paper in the introduction.

3. The discussion may consider a deeper connection between the empirical results of this paper and existing research findings.

4. Highlight the applicability of the results of this paper in the conclusion.

5. In 3.2 mentions "five casual tourist interviews were conducted in this process ", the article is mainly through UGC data, whether the interviews with just five visitors appeared in the paper is meaningful.

6. The map should be printed on a base map published by the Ministry of Natural Resources, and in addition, the colors could be changed to a more contrasting color as in Figure 9.

There's no problem with the language.

Author Response

Thank you for the feedback on this study. The manuscript is much improved based on the suggestions you have made. We hope this revision meets your satisfaction. The point-to-point responses to your comments are in the attached file.

Reviewer 2 Report

Thanks for the opportunity to review this interesting study. The manuscript is in good shape overall. Research results are supported with multiple quantitative and qualitative studies, and presented with rich figures and tables. 

Good quality overall.

Author Response

Thank you for the feedback on this study. And thanks for your recognition of this manuscript.

Reviewer 3 Report

The studies were well planned, and best possible parameters were implemented to determine the impacts of climate change on Water Heritage Tourism in China and then the role of relevant stockholders to cope with the impacts of climate change. The results have been well presented, however I have some suggestions for improvements (in the attached file.)

Slight language improvement required

Author Response

Thank you for your feedback on this study. The manuscript is much improved based on the suggestions you have made. We hope this revision meets your satisfaction. And the point-to-point responses to your comments are in the attached file.

Reviewer 4 Report

Reviewer Report

Research on Climate Change and Water Heritage Tourism based on the Adaptation Theory —A Case Study of the Grand Canal (Beijing Section)

3. Materials and Methods

Data Collection and Processing

Reviewer: The process of collection and analysis are not clear, it is necessary to detail the process and explain for example what adaptations were made, what previous studies validate the choices of method and software used, what questionnaire was used, if it was adapted what scales were used.

5. Discussion and Implications

Reviewer: The discussion should be done with previous literature, i.e., show how the results of this study are in line with or differ from previous studies.

6. Conclusions

Reviewer: The authors should reinforce contributions to the study, it would be important to mention what gap or gaps found in the literature are answered in this study. In this way, the article would make a greater scientific contribution.

The English language, needs to be reviewed, ideally by a native. 

Author Response

(The authors gave the same response as above.)
